# The Effects of Breastfeeding on Retinoblastoma Development: Results from an International Multicenter Retinoblastoma Survey

**DOI:** 10.3390/cancers13194773

**Published:** 2021-09-24

**Authors:** Jasmeen K. Randhawa, Mary E. Kim, Ashley Polski, Mark W. Reid, Kristen Mascarenhas, Brianne Brown, Ido Didi Fabian, Swathi Kaliki, Andrew W. Stacey, Elizabeth Burner, Caitlin S. Sayegh, Roy A. Poblete, Xunda Ji, Yihua Zou, Sadia Sultana, Riffat Rashid, Sadik Taju Sherief, Nathalie Cassoux, Juan Garcia, Rosdali Diaz Coronado, Arturo Manuel Zapata López, Tatiana Ushakova, Vladimir G. Polyakov, Soma Rani Roy, Alia Ahmad, M. Ashwin Reddy, Mandeep S. Sagoo, Lamis Al Harby, Nicholas John Astbury, Covadonga Bascaran, Sharon Blum, Richard Bowman, Matthew J. Burton, Nir Gomel, Naama Keren-Froim, Shiran Madgar, Marcia Zondervan, Jesse L. Berry

**Affiliations:** 1The Vision Center at Children’s Hospital Los Angeles, Los Angeles, CA 90027, USA; jkrandha@usc.edu (J.K.R.); maryekim@usc.edu (M.E.K.); ashley.polski@usc.edu (A.P.); mreid@chla.usc.edu (M.W.R.); bribrown@chla.usc.edu (B.B.); 2USC Roski Eye Institute, Keck School of Medicine, University of Southern California, Los Angeles, CA 90033, USA; 3Miller School of Medicine, University of Miami, Miami, FL 33136, USA; kmascarenhas@med.miami.edu; 4International Centre for Eye Health, London School of Hygiene and Tropical Medicine, London WC1E 7HT, UK; didi@didifabian.com (I.D.F.); nick.astbury@lshtm.ac.uk (N.J.A.); covadonga.bascaran@lshtm.ac.uk (C.B.); richardbowman493@gmail.com (R.B.); matthew.burton@lshtm.ac.uk (M.J.B.); marcia.zondervan@lshtm.ac.uk (M.Z.); 5Sheba Medical Center, Goldschleger Eye Institute, Tel Hashomer, Tel-Aviv University, Tel-Aviv 52621, Israel; sharon.blum@gmail.com (S.B.); naamake69@gmail.com (N.K.-F.); shiran.madgar@gmail.com (S.M.); 6The Operation Eyesight Universal Institute for Eye Cancer, Hyderabad 500034, India; kalikiswathi@yahoo.com; 7Department of Ophthalmology, University of Washington, Seattle, WA 98195, USA; andrewstacey@gmail.com; 8Department of Emergency Medicine, Keck School of Medicine, University of Southern California, Los Angeles, CA 90033, USA; eburner@usc.edu; 9Department of Pediatrics, Keck School of Medicine, University of Southern California, Los Angeles, CA 90033, USA; caitlias@usc.edu; 10Department of Neurology, Keck School of Medicine, University of Southern California, Los Angeles, CA 90033, USA; rpoblete@usc.edu; 11Department of Ophthalmology, Xinhua Hospital, Shanghai Jiao Tong University School of Medicine, Shanghai 200025, China; jixunda2007@aliyun.com (X.J.); 15221832276@163.com (Y.Z.); 12Department of Oculoplasty and Ocular Oncology, Ispahani Islamia Eye Institute and Hospital, Dhaka 1215, Bangladesh; sadia.sultana@islamia.org.bd (S.S.); drriffatrashid@gmail.com (R.R.); 13Department of Ophthalmology, School of Medicine, Addis Ababa University, Addis Ababa 3614, Ethiopia; goge4000@yahoo.com; 14Institut Curie, Université de Paris, 75248 Paris, France; nathalie.cassoux@curie.fr; 15Clinica Angloamericana, Lima 15073, Peru; jgarcia@inen.sld.pe; 16Instituto Nacional de Enfermedades Neoplasicas, Lima 15038, Peru; rosdali.diaz.c@upch.pe (R.D.C.); zapatalopezarturo@gmail.com (A.M.Z.L.); 17N.N. Blokhin National Medical Research Center, Head and Neck Tumors Department, SRI of Pediatric Oncology and Hematology, Oncology of Russian Federation, 115478 Moscow, Russia; ushtat07@mail.ru (T.U.); vgp-04@mail.ru (V.G.P.); 18Medical Academy of Postgraduate Education, 125445 Moscow, Russia; 19Chittagong Eye Infirmary & Training Complex, Chittagong 4202, Bangladesh; dr.somaroy2020@gmail.com; 20The Children’s Hospital and the Institute of Child Health, Lahore 54000, Pakistan; alia.ahmad@stjude.org; 21The Royal London Hospital, Barts Health NHS Trust, London E1 1BB, UK; ashwin.reddy4@nhs.net (M.A.R.); mandeep.sagoo1@nhs.net (M.S.S.); lamis.alharby1@nhs.net (L.A.H.); 22Moorfields Eye Hospital NHS Foundation Trust, London EC1V 2PD, UK; 23UCL Institute of Ophthalmology, London EC1V 2PD, UK; 24Ophthalmology Department, Great Ormond Street Children’s Hospital, London WC1N 3JH, UK; 25Tel Aviv Sourasky Medical Center, Division of Ophthalmology, Sackler Faculty of Medicine, Tel-Aviv University, Tel-Aviv 39040, Israel; nir.gomel1@gmail.com; 26Norris Comprehensive Cancer Center, Keck School of Medicine, University of Southern California, Los Angeles, CA 90033, USA; 27The Saban Research Institute, Children’s Hospital Los Angeles, Los Angeles, CA 90027, USA

**Keywords:** eye, tumor, pediatric cancer, retinoblastoma, breastfeeding

## Abstract

**Simple Summary:**

Breastfeeding has been shown to lower the risk of oncogenesis in many pediatric cancers, with longer periods of breastfeeding having the most protective effect. However, an association has not yet been determined for the consequence or benefit of breastfeeding in retinoblastoma (RB), the most common intraocular malignancy of childhood affecting 8000 children worldwide each year. Herein, we aimed to understand the role of breastfeeding in the severity of development of nonhereditary RB, specifically its relationship to age at diagnosis, ocular prognosis, and extraocular involvement. Our analysis of 344 patients indicated that neither breastfeeding nor formula feeding was associated with differences in age at diagnosis, ocular prognosis, or extraocular involvement. More research elucidating the factors affecting the development of RB is warranted both to understand the pathophysiology of tumor development and to develop clinical recommendations for preventive care.

**Abstract:**

The protective effects of breastfeeding on various childhood malignancies have been established but an association has not yet been determined for retinoblastoma (RB). We aimed to further investigate the role of breastfeeding in the severity of nonhereditary RB development, assessing relationship to (1) age at diagnosis, (2) ocular prognosis, measured by International Intraocular RB Classification (IIRC) or Intraocular Classification of RB (ICRB) group and success of eye salvage, and (3) extraocular involvement. Analyses were performed on a global dataset subgroup of 344 RB patients whose legal guardian(s) consented to answer a neonatal questionnaire. Patients with undetermined or mixed feeding history, family history of RB, or sporadic bilateral RB were excluded. There was no statistically significant difference between breastfed and formula-fed groups in (1) age at diagnosis (*p* = 0.20), (2) ocular prognosis measures of IIRC/ICRB group (*p* = 0.62) and success of eye salvage (*p* = 0.16), or (3) extraocular involvement shown by International Retinoblastoma Staging System (IRSS) at presentation (*p* = 0.74), lymph node involvement (*p* = 0.20), and distant metastases (*p* = 0.37). This study suggests that breastfeeding neither impacts the sporadic development nor is associated with a decrease in the severity of nonhereditary RB as measured by age at diagnosis, stage of disease, ocular prognosis, and extraocular spread. A further exploration into the impact of diet on children who develop RB is warranted.

## 1. Introduction

Retinoblastoma (RB) is a cancer that forms in one or both eyes of infants and toddlers. Although rare, RB is the most common pediatric ocular malignancy, affecting 8000 children worldwide each year [1]. In the vast majority of cases, both alleles of the *RB1* gene acquire a loss of function mutation in order for a child to develop RB. This can occur in either a hereditary or nonhereditary pattern. In hereditary RB, the first mutation arises in the germline, either de novo or inherited from a parent, and thus is present in all cells of the body. Tumorigenesis occurs when a second hit on the *RB1* gene is acquired in any retina cell. Forty percent of RB patients have this genetic predisposition; these patients more commonly present at younger ages and with bilateral disease affecting both eyes. The other 60% of patients have a nonhereditary form of RB, wherein two de novo somatic events occur within a single retina cell. Children with nonhereditary RB present with unifocal disease in only one eye.

The hereditary genetic mechanisms underlying RB tumorigenesis are well characterized, but the role of perinatal nutrition and parental health factors in the etiology of developing new somatic mutations are incompletely understood and the current literature on the subject is conflicting. Maternal use of multivitamins [2] and condom use [3] (likely related to decreased human papilloma virus infection) may lower risk of germline RB mutations. Maternal smoking before and during pregnancy [4] and higher levels of gestational intake of cured meats [5] are associated with an increased risk of the child developing nonhereditary unilateral RB. A maternal diet containing more fruits [5] and vegetables [6] may reduce this risk. Poor paternal diet [7], low maternal weight [3], and maternal prescription pain medication use [3] are also linked to hereditary RB development. However, the role of breastfeeding in the development of somatic RB mutations is unknown. In other pediatric malignancies, breastfeeding has been shown to lower the risk of oncogenesis, with longer periods of breastfeeding having the most protective effect [8,9,10,11]. Breastmilk may help stimulate and promote development of the infant immune system. These strengthened immune systems can then dampen the effects of infectious exposures, decreasing spontaneous mutations, which lead to certain cancers [9].

While the protective effects of breastfeeding on various childhood malignancies have been investigated, the role of breastfeeding in the development of RB is unclear. There is limited data on RB incidence among breastfed and formula-fed children and an absence of data on retinoma formation, which if available may provide even better insight into the possible protective effect of breastfeeding. A multicenter case-control study assessing 282 cases of RB found that breastfeeding for seven to eleven months was associated with decreased risk of developing nonhereditary RB, but no dose-response effect was observed for longer durations of breastfeeding [3]. Conversely, a group in Sweden found that prolonged breastfeeding was predictive of a nonsignificant increased risk of RB development [12]. Given these inconclusive results, we aim to further investigate the role breastfeeding plays in the severity of nonhereditary RB development. Specifically, we will assess the relationship of breastfeeding to (1) patient age at diagnosis, (2) ocular prognosis, and (3) extraocular involvement using a large published multinational database [13]. We limited our investigation to nonhereditary cases of RB, as infant nutrition is less likely to influence the development of hereditary RB given that the first mutation occurs during embryogenesis before the infant breastfeeds. We hypothesized that maternal breastfeeding may yield protection from the development of nonhereditary RB, resulting in (1) older patient age at diagnosis, (2) improved ocular prognosis, and (3) less extraocular extension. As a secondary analysis, we also assessed the effects of socioeconomic status (SES) at the nation level on these outcomes, as prior research has also suggested a role in the severity of RB [14,15,16,17,18] as well as a complex role in the rates of breastfeeding [19,20,21,22].

## 2. Materials and Methods

This study is a secondary database analysis completed on a deidentified dataset from patients who participated in a multinational collaborative study, the Lag Time for RB study [13]. The Lag Time study was a collaboration of 11 RB treatment centers and was approved by the London School of Hygiene & Tropical Medicine Institutional Review Board (reference no. 15882). Informed consent was obtained from all parents/guardians of the children included in this study. Each center received local approval according to their institutional guidelines. Participating centers included Addis Ababa (Ethiopia), Hyderabad (India), Dhaka and Chittagong (both from Bangladesh), Lahore (Pakistan), Lima (Peru), Moscow (Russia), Shanghai (China), Los Angeles (USA), London (UK), and Paris (France). At our site, this study was approved by the Children’s Hospital Los Angeles Institutional Review Board and conformed to the tenets of the Declaration of Helsinki.

### 2.1. Inclusion Criteria

Inclusion criteria for our subgroup analysis were patients who participated in the Lag Time for RB study [13] who completed the neonatal portion of the questionnaire. Parents were asked to indicate breastfeeding status as “yes”, “milk substitute”, “both”, or “other”.

### 2.2. Exclusion Criteria

Patients who reported “both” or “other” feeding status on the neonatal questionnaire were excluded from this secondary analysis. Patients with a family history of RB or who developed bilateral RB during their time in the study were excluded from analyses. This included two patients with missing data for final laterality. Some centers did not use any group classification; patients from these centers were excluded from the analysis of stage at presentation but were not excluded from other analyses.

### 2.3. Statistical Analysis

Categorical demographic and clinical variables were compared between breastfed and formula-fed patients using Chi-squared tests or Fisher’s exact tests, where appropriate. Age at diagnosis was compared between these groups using an independent samples *t*-test. To examine the effect of breastfeeding status on clinical outcomes, including IIRC/ICRB and IRSS stages at presentation, enucleation, lymph node involvement, distant metastases, and death, a series of logistic regression models without and with covariates were used on each outcome. First, a univariate logistic regression model was used to examine the relative odds of each outcome in breastfed patients compared to formula-fed patients. Next, a second logistic regression model was used to examine the independent effect of breastfeeding on this outcome while accounting for age and dichotomized SES as covariates. Wilk’s likelihood-ratio test was used to evaluate the change in significance between this second model and one with only age and SES predicting an outcome. Dichotomous outcomes, including enucleation, lymph node involvement, distant metastases, and death (yes or no) were evaluated using binary logistic regression models. Both IIRC/ICRB and IRSS stages at presentation were evaluated using ordered logistic regression models, otherwise following the same procedure. IIRC/ICRB stage at presentation was also evaluated dichotomously as low risk (0, Groups A–C) or high risk (1, Groups D and E) based on either grouping system, which primarily differed in their definitions of high risk groups D and E [23]. Independent effects of dichotomized SES on enucleation, lymph node involvement, distant metastases, and death were evaluated using binary logistic regression models; effects on age at diagnosis were evaluated using ordinary least squares regression; effects on IIRC/ICRB stage at presentation were evaluated using ordered logistic regression. All *p*-values of <0.05 were considered statistically significant. All analyses were conducted using Stata/SE 14.2 (StataCorp LLC, College Station, TX, USA). Posthoc power analyses were done with GPower 3.1 (University of Southern California, Los Angeles, CA, USA).

## 3. Results

A retrospective subgroup analysis was performed on treatment-naïve RB patients who presented to the participating centers from 1 January 2019 to 31 December 2019. The inclusion criteria were met in 654 patients of 692 patients who were included in the Lag Time study (94.5%). The neonatal section was completed if the physician was able to elicit a complete feeding history from the patient. One hundred and twenty patients noted “combination” or “other” breastfeeding history; these patients were excluded from the analysis because no detailed descriptions of their feeding behaviors were provided, and thus the variety of feeding behaviors could not be stratified. Of the remaining patients, 19 and 171 hereditary RB patients were excluded based on having a family history of RB or bilateral RB, respectively. Ultimately, 344 patients were included in the analysis. Of the 344 patients in our analysis, 293 (85.2%) patients were exclusively breastfed. Additional patient demographics are shown in Table 1.

### 3.1. Breastfeeding and Age at Diagnosis

There was no significant difference in the age at diagnosis between breastfed and formula-fed patients in an independent samples *t*-test (*p* = 0.20). Breastfed patients were diagnosed at a mean of 28.2 ± 18.8 months, while formula-fed patients were diagnosed at a mean of 24.6 ± 13.3 months (Table 1).

### 3.2. Breastfeeding and Ocular Prognosis

The large majority of breastfed patients were classified as IIRC/ICRB Group D (*n* = 75, 25.6%) or Group E (*n* = 170, 58.0%), and 158 (51.5%) children required enucleation as treatment for RB. The majority of formula-fed patients were also classified as Group D (*n* = 19, 37.3%) and Group E (*n* = 28, 54.9%), with 19 (37.3%) formula-fed children requiring enucleation. The distribution of IIRC/ICRB group at presentation and enucleation status by feeding type are displayed in Table 2.

When IIRC/ICRB stage at presentation [23] was dichotomized as low-risk intraocular disease (Groups A, B, or C) versus high-risk (Groups D and E), those in the formula feeding group did not have significantly greater odds of presenting in the high risk-group than those in the breastfeeding group (OR = 1.12, *p* = 0.85, 95% CI = 0.36, 3.51) after controlling for age and SES. Similarly, in ordered logistic regression, breastfeeding did not reduce the severity of RB at presentation as measured by the IIRC/ICRB Group (OR = 0.86, *p* = 0.62, 95% CI = 0.48, 1.56), after controlling for age and SES.

Univariate logistic regression analysis indicated an increased odds of enucleation for infants exclusively breastfed compared to those who were formula-fed (OR = 1.97, *p* = 0.03, 95% CI 1.07, 3.64). This increased risk was found to be due to confounding, as the estimated coefficient for breastfeeding and enucleation decreased by more than 15% when SES of the patients’ countries was added to the model. This increased risk for enucleation for breastfed patients was not significant after controlling for the patient’s age and the SES of the country where the patient was treated (OR = 1.58, *p* = 0.16, 95% CI = 0.84, 3.00). The multivariable model did not demonstrate significantly greater fit when breastfeeding status was added (χ^2^ (1) = 2.01, *p* = 0.16; Table 3).

### 3.3. Breastfeeding and Extraocular Involvement

Six (2.0%) breastfed patients had regional lymph node involvement while 23 (7.8%) demonstrated metastatic disease. In contrast, none of the formula-fed patients had regional lymph node involvement, and only one (2.0%) formula-fed patient suffered metastatic disease. Ten (3.9%) breastfed patients died, while no formula-fed patients died during the duration of the study. The distribution of extraocular involvement and breastfeeding status is displayed in Table 4. Although being exclusively breastfed was associated with a significantly greater odds of presenting with a higher IRSS stage (OR = 1.94, *p* = 0.03, 95% CI = 1.08, 3.50), this association was not significant after accounting for age and SES (OR = 1.11, *p* = 0.74, 95% CI = 0.60, 2.05). There were no differences in lymph node involvement (OR = 0.19, *p* = 0.20, 95% CI = 0.01, 2.41) or distant metastases (OR = 2.60, *p* = 0.37, 95% CI = 0.33, 20.75) between RB patients who were breastfed and those who were formula-fed in multivariable models controlling for age and SES. Because no patients in the formula-fed group died, the effects of breastfeeding on odds of death could not be evaluated.

### 3.4. Effect of SES on RB Outcomes

Breastfed patients were more likely to be from a low- or lower-middle-income country, while formula-fed patients were more likely to be from an upper-middle- or high-income country (*p* < 0.0001; Table 1). Consistent with previous studies, patients from upper-middle- or high-income countries were generally diagnosed at younger ages (B = −6.26 months, *p* = 0.001, 95% CI = −10.09, −2.43), were less likely to require enucleation (OR = 0.52, *p* = 0.003, 95% CI = 0.34, 0.80), and suffered less distant metastases (OR = 0.26, *p* = 0.02, 95% CI = 0.09, 0.77) than patients from low- or lower-middle-income countries. In addition, those patients from upper-middle- or high-income countries showed lower odds of lymph node involvement (OR = 0.08, *p* = 0.09, 95% CI = 0.005, 1.44) and lower odds of death (OR = 0.29, *p* = 0.12, 95% CI = 0.06, 1.39) compared to those from low- and lower-middle-income countries, although these associations were not statistically significant.

### 3.5. Posthoc Power Analysis

Using G *Power, we performed a posthoc power analysis to find the sample size required to observe a significant difference in odds of enucleation based on feeding status after accounting for covariates. We used a *z*-test specifying a binomial distribution and subgroup size (breast vs. formula feeding), with an R2 of 0.02 for other predictors effect on enucleation. Our study showed a 2-tailed posthoc power of 56% in detecting a difference in odds of enucleation between the groups, suggesting data on 596 patients would have demonstrated a significant independent effect of breastfeeding status.

## 4. Discussion

In this multicenter, multinational analysis of patients with RB, there was no significant association observed between breastfeeding and the severity of nonhereditary RB development. These findings are inconsistent with what was hypothesized based on prior literature, which has suggested that breastfeeding provides a protective role in the development of various cancers due to antimicrobial, anti-inflammatory, and immunomodulating effects [24]. The recognition of breastfeeding as the healthiest way to feed an infant [25] has spurred the creation of breastfeeding initiatives in the United States, where the Surgeon General’s Call to Action to Support Breastfeeding actively discourages formula-feeding and encourages providing care for lactating mothers [26]. In multiple studies, breastfeeding was associated with a lower risk of oncogenesis for a variety of pediatric malignancies [8,9,10,11]. For example, a study investigating 300 patients with various pediatric cancers including RB, breastfeeding lowered the risk of cancer development regardless of cancer type [10]. The mechanism for protection against cancer is not entirely understood, but it is thought to involve specific substances such as human soluble tumor necrosis factor-related apoptosis-inducing ligand [27] and human alpha-lactalbumin [28]; these help control cell proliferation across the body and cause apoptosis in tumor cells.

In this study, patients from higher SES countries were significantly more likely to utilize formula-feeding. This may be due to greater accessibility or affordability of formula, or the convenience of formula if the mother is employed. A mothers’ choice to breastfeed over formula-feed is inherently intertwined with multiple facets of her SES including education [29], employment [29], income [21], as well as cultural and societal values. Prior inquiries into understanding this decision-making process have led to variable results. Heck and colleagues observed that women with higher education levels are more likely to breastfeed in the United States, with income or occupation no longer significant once adjusted for confounders [21]. However, their investigation has also shown that despite lower education and family incomes, foreign-born Latina women were most likely to breastfeed out of any other group in the United States [21]. As our investigation of SES focused on comparison between nations, there may be additional unmeasured effects of SES contributing to the choice of breastfeeding over formula feeding in participating mothers.

Our study showed that patients with a higher SES were less likely to necessitate enucleation, indicating better ocular prognosis. Numerous studies have noted the challenges that patients of lower SES face, including greater distance to a treatment center, more comorbidities, lack of knowledge of early signs of RB, and fewer treatment options available; these obstacles lead to advanced disease at presentation and result in poorer outcomes [30]. Developed nations show a two to five percent rate of advanced disease in cases of RB, whereas developing nations show a 30–40% rate of advanced disease in cases of RB [30,31,32]. This disparity leads to a 50% survival of children with RB worldwide in contrast to the survival rate of >90% in resource-rich countries [31]. It is important to note that low SES does not result in RB [33,34]. Rather, a patient of low SES suffers decreased access to resources that usually result in expedient diagnosis, treatment, and cure for patients of high SES. In this study, it is likely that the multifaceted benefits a patient experienced due to higher SES had a more direct, causal link than any benefit that could have been conferred to a patient through the potentially immunomodulatory effects of breastfeeding alone.

There were a variety of limitations in this study, both due to the reporting of family history and constraints of the sample size. While data on family history of RB were noted by many participating centers, these data were not explicitly collected and therefore may be incomplete, resulting in an underestimation of the number of familial cases in our dataset. Development of RB in these missed familial cases would not have been impacted by postnatal feeding and thus may confound the results. While this is one of the largest multinational clinical databases on retinoblastoma, some patient reports were incomplete and lacked data in certain categories (such as lymph node involvement), which may also impact the analysis. Patients were followed for one year, so there is a possibility that death or metastasis could have occurred after the conclusion of the followup period. Another limitation of this study is that one cannot determine whether an initial somatic mutation occurred during the prenatal period, which would not have been affected by diet. Lastly, an a priori power calculation was not feasible for this study due to the lack of significant previous data on this topic to evaluate effect as well as the secondary database analysis nature of the study. Therefore, it was uncertain prior to analysis if our sample size was adequate to elucidate an effect based on feeding status. While the number of patients included in this study is comparable to the sample size of other studies on nutritional impact on RB, we were powered to detect a 56% difference in outcomes between the groups given the overwhelming majority of patients who were breastfed and the large contribution of covariates. Based on our cohort demographics, we could only detect a significant odds ratio of 2.4 or higher for breastfeeding as a risk factor for enucleation. Ideally, future data about breastfeeding and formula feeding habits would be collected within one country with similar SES amongst its citizens to preserve an effect if one exists. However, this is a challenge for studying rare diseases such as RB: in the United States, less than 300 children are diagnosed annually [35].

## 5. Conclusions

This multicenter international study has not demonstrated a relationship between breastfeeding and the severity of nonhereditary RB development. An understanding of the environmental factors contributing to sporadic development of RB is the first step in creating appropriate clinical recommendations for new parents looking to optimize their child’s peri- and postnatal nutrition. This study contradicts the few reports of both negative and positive associations of breastfeeding with RB severity, and therefore provides an impetus for further research on this topic. Previously published findings stating children from higher SES countries have better outcomes were confirmed in adjunct analyses; because of this, addressing the disparities in care across various socioeconomic backgrounds remains a top priority. More work is needed to elucidate the factors affecting the development of RB, both to understand the pathophysiology of tumorigenesis and to improve clinical recommendations for preventative care.

## Figures and Tables

**Table 1 cancers-13-04773-t001:** Distribution of demographics and breastfeeding status.

Demographics	Total *N* (Patients) = 344		Breastfed *n* = 293		Formula-Fed *n* = 51		*p* Value ^1^
	*n*	%	*n*	%	*n*	%	
Gender							0.13
Female	155	45.1	127	43.3	28	54.9	
Male	189	54.9	166	56.7	23	45.1	
Age at Diagnosis (mos) ^2^							0.20
Mean	27.6		28.2		24.6		
Std Dev ^2^	18.1		18.8		13.3		
Economic Grouping (SES) ^2^							<0.0001
Low/Low Middle	197	57.3	185	63.1	12	23.5	
Upper Middle/High	147	42.7	108	36.9	39	76.5	
Term status							0.60
Full-term	278	80.8	236	80.5	42	82.4	
Pre-term	66	19.2	57	19.5	9	17.6	
Birth Order							0.77
First born	142	41.3	120	40.9	22	43.1	
Not first born	202	58.7	173	59.0	29	56.9	

^1^ Frequencies of patients across demographic categories were compared using Chi-squared tests; age at diagnosis was compared between breastfed and formula-fed patients using an independent samples *t*-test. ^2^ Abbreviations: Mos: months. Std Dev: Standard deviation. SES: Socioeconomic status.

**Table 2 cancers-13-04773-t002:** Distribution of ocular prognosis measures and breastfeeding status.

Ocular Prognosis Measures	Total *N* (Patients) = 344		Breastfed *n* = 293		Formula-Fed *n* = 51		*p* Value
	*n*	%	*n*	%	*n*	%	
Group ^1^							0.36
A	1	0.3	1	0.3	0	0	
B	2	0.6	2	0.7	0	0	
C	6	1.7	5	1.7	1	2.0	
D	94	27.3	75	25.6	19	37.3	
E	198	57.6	170	58.0	28	54.9	
Unknown Group	43	12.5	40	13.7	3	5.9	
Enucleation Status							0.03
Yes	177	51.5	158	53.9	19	37.3	
No	167	48.6	135	46.1	32	62.7	

^1^ Since data was collected globally, both IIRC and ICRB classifications were used according to the standard of each respective clinic. Classifications are combined in this table.

**Table 3 cancers-13-04773-t003:** Univariate and multivariable model statistics for enucleation.

Univariate and Multivariable Models	Dependent Variable = Enucleation		
	OR	95% CI	*p* Value
Univariate Model ^1^			
Breastfeeding	1.97	1.07, 3.64	0.03
Log Likelihood	−235.86		
Multivariable Model ^2^			
Breastfeeding	1.58	0.84, 3.00	0.16
Socioeconomic status	0.58	0.37, 0.92	0.02
Age	1.00	0.99, 1.02	0.53
Log Likelihood	−232.17		
Covariate Model ^2^			
Socioeconomic status	0.53	0.34, 0.83	0.005
Age	1.00	0.99, 1.02	0.51
Log Likelihood	−233.64		

^1^ Univariate model pseudo-R^2^, 0.01. ^2^ Multivariable and Covariate model pseudo-R^2^, 0.02.

**Table 4 cancers-13-04773-t004:** Distribution of extraocular measures and breastfeeding status.

Extraocular Prognosis Measures	Total *N* (Patients) = 344		Breastfed *n* = 293		Formula-Fed *n* = 51		*p* Value
	*n*	%	*n*	%	*n*	%	
Lymph Node Involvement							1.00
No involvement	149	83.2	143	83.1	6	85.7	
Evidence of regional involvement	6	3.4	6	3.5	0	0.0	
Distant Metastases	24	13.4	23	13.4	1	14.3	
Not assessed	165	−	121	−	44	-	
IRSS ^1^ Stage							0.30
0	136	41.9	108	39.1	28	57.1	
I	96	29.5	85	30.8	11	22.4	
II	60	18.5	54	19.6	6	12.2	
IIIa	13	4.0	10	3.6	3	6.1	
IIIb	4	1.2	4	1.5	0	0.0	
IVa	2	0.6	2	0.7	0	0.0	
IVb	14	4.3	13	4.7	1	2.0	
Unknown	19	−	17	−	2	−	
Death							0.37
Yes	10	3.2	10	3.8	0	0.0	
No	305	96.8	255	96.2	50	100.0	
Unknown	29	−	28	−	1	−	

^1^ Abbreviations: IRSS: International Retinoblastoma Staging System.

## Data Availability

Data presented in this study are available on request from the corresponding author. The data are not publicly available due to ethical concerns.

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
