# Peer review of "The Effects of Breastfeeding on Retinoblastoma Development: Results from an International Multicenter Retinoblastoma Survey"

_cancers, 2021, doi:10.3390/cancers13194773_

Round 1

Reviewer 1 Report

Randhawa, Berry and colleagues report interesting results of a multicenter, multinational study on the influence of breasfeeding on the occurrence of retinoblastoma.

The findings heavily rely on strong statistical calculations and models, and convincingly show with a necessarily limited number of cases due to the rarity of the condition, that breastfeeding has no influence on retinoblastoma development. The consideration of country SES, and the inclusion of treatment data are critical to warrant the robustness of the analysis.

Importantly, this type of study relying on heterogeneous clinical data, and simplifying mathematical models, inherently presents several flaws, clearly identified and minimized by the authors in the Results and Discussion sections.

Authors should address the following minor suggestions:

Table 4 : IRSS Stages IVa and IVb are misspelled

Discussion line 290: why is Ref 10 only quoted among studies on breastfeeding and childhood cancer occurence?  Is it prospective? Is so, authors shoud specify, or rephrase to mention the major studies on the topic (as mentioned in the introduction).

Authors should add to the Discussion on study limitations the possibility of pre natal somatic mutation occurrence, inducing cases of sporadic Rb at younger age, that the breasfeeding status would not influence.

Authors should also mention the absence of data on retinoma incidence among breastfed and formula-fed children, which would provide a useful insight on the possible protective status of breastfeeding.

The mention that ‘All participating centers 380 received clearance from their respective institutional review board and ethics committee for partic- 381 ipating in this international collaborative study’ should be added to the Methods section, in addition to the ethical approval from the coordinating centre.

Author Response

Reviewer 1

  1. Table 4: IRSS Stages IVa and IVb are misspelled

Thank you for taking the time to review our paper and provide valuable feedback. We have fixed this error.

  1. Discussion line 290: why is Ref 10 only quoted among studies on breastfeeding and childhood cancer occurrence?  Is it prospective? Is so, authors should specify, or rephrase to mention the major studies on the topic (as mentioned in the introduction).

Thank you for this comment. Reference 10 was a retrospective case-control study. We have added additional language summarizing the points from the intro with their citations: In multiple studies, breastfeeding was associated with a lower risk of oncogenesis for a variety of pediatric malignancies [8-11]. For example, a study investigating 300 patients with various pediatric cancers including RB, breastfeeding lowered the risk of cancer development regardless of cancer type [10]. (Line 304).

  1. Authors should add to the Discussion on study limitations the possibility of prenatal somatic mutation occurrence, inducing cases of sporadic Rb at younger age, that the breastfeeding status would not influence.

Thank you for your comment. On line 357, we have added the following sentence: Another limitation of this study is that one cannot determine whether an initial somatic mutation occurred during the prenatal period, which would not have been affected by diet.

  1. Authors should also mention the absence of data on retinoma incidence among breastfed and formula-fed children, which would provide a useful insight on the possible protective status of breastfeeding.

Thank you for your comment. We have mentioned the absence of RB data described with breastfed/formula-feeding status as one of the reasons for conducting the study in the introduction; there is an absence of data on retinomas as the reviewer points out. These lesions are so rare that likely there will never be enough data on retinomas to make a conclusive statement here, and not all readers may understand the differences between RB and retinoma in the same way the reviewers clearly does, however we have added a general statement to address this for those that do.

On line 127 we have added the following sentence to state: There is limited data on RB incidence among breastfed and formula-fed children and an absence of data on retinoma formation, which if available may provide even better insight into the possible protective effect of breastfeeding..

  1. The mention that ‘All participating centers received clearance from their respective institutional review board and ethics committee for participating in this international collaborative study’ should be added to the Methods section, in addition to the ethical approval from the coordinating centre.

Thank you for this comment. The methods section includes a statement that says “The Lag Time study was a collaboration of 11 RB treatment centers and was approved by the London School of Hygiene & Tropical Medicine Institutional Review Board (reference no. 15882). Informed consent was obtained from all parents/guardians of the children included in this study. Each center received local approval according to their institutional guidelines. Participating centers included Addis Ababa (Ethiopia), Hyderabad (India), Dhaka and Chittagong (both from Bangladesh), Lahore (Pakistan), Lima (Peru), Moscow (Russia), Shanghai (China), Los Angeles (USA), London (UK) and Paris (France). At our site, this study was approved by the Children’s Hospital Los Angeles Institutional Review Board and conformed to the tenets of the Declaration of Helsinki.”

Reviewer 2 Report

The manuscript investigates the association between breastfeeding and characteristics including ocular prognosis measures, age at diagnosis, and extraocular involvement among 344 patients with non-hereditary retinoblastoma. The study population is a subset of a larger international study, the Lag Time for Rb Study. After adjusting for socioeconomic status, the authors found no association between breastfeeding and any of the Rb characteristics examined.   

Overall, this is a well-written manuscript. In particular, the authors do a very comprehensive job in the analysis and discussion addressing socioeconomic status, which is associated with both breastfeeding and stage at RB diagnosis and thus an important confounder.

Main comments and questions:

  1. My main comment concerns the hypothesis: “Maternal breastfeeding may yield protection against the development of non-hereditary RB, resulting in (1) older patient age at diagnosis, (2) improved ocular prognosis, and (3) less extraocular extension.”

Understanding modifiable factors associated with stage and severity can have implications for improving survival in areas of the world where Rb is often diagnosed at advanced stages. This analysis clearly examines the association between breastfeeding and disease severity. However, it is not clear to me how these relate back to the risk for developing non-hereditary Rb, or even how the question of risk for developing Rb can be investigated in a study population where all patients have the outcome.

  1. Related, the authors state in the Abstract and Discussion that they found “no significant protective role of breastfeeding… in the development of nonhereditary RB.” It would seem more accurate to say no association was observed between breastfeeding and severity of disease as measured by …..

  1. From the original Lag study, what proportion had the necessary data (i.e., completed the neonatal section)? What factors were associated with completion of this section?

  1. How long were patients followed to determine whether they had died? Whereas other analyses pertain to characteristics around the time of diagnosis or shortly after (e.g., enucleation), mortality requires patient follow-up. More information about follow-up methods and completeness is needed to understand this analysis.

  1. For some factors (e.g., lymph node involvement) there is considerable missing data. Can the authors discuss the potential implications of the missing data?

Additional comments:

  1. How strong is the evidence from previous literature regarding perinatal and nutritional factors in terms of RB etiology? In its current wording, the introduction seems to suggest that the evidence is quite strong, but when I reviewed the publications cited in this section, it does not seem so clear.
  2. Please clarify how misclassification of hereditary status due to missing family history information could confound the study results (page 8, line 327).

Author Response

Reviewer #2

  1. My main comment concerns the hypothesis: “Maternal breastfeeding may yield protection against the development of non-hereditary RB, resulting in (1) older patient age at diagnosis, (2) improved ocular prognosis, and (3) less extraocular extension.”

Understanding modifiable factors associated with stage and severity can have implications for improving survival in areas of the world where Rb is often diagnosed at advanced stages. This analysis clearly examines the association between breastfeeding and disease severity. However, it is not clear to me how these relate back to the risk for developing non-hereditary Rb, or even how the question of risk for developing Rb can be investigated in a study population where all patients have the outcome.

Thank you for taking the time to review our paper and provide valuable feedback. You are correct in saying that studying the risk of developing RB is better suited to a study that includes patients who also do not have the outcome. By comparing formula-fed and breastfed groups of those who developed RB, we aimed to understand how feeding affects development of RB through disease severity, not a risk assessment of whether one would develop RB or not. We believed the immune modulating effects of breast milk could lead to a lesser burden of disease in those who did develop disease. We can see how this language lacks clarity, so we have adjusted it within the paper on lines 68, 77, 135, 141, 297, and 374 which now define our aims as assessing the severity of development. Thank you for bringing up this important clarifying point.

  1. Related, the authors state in the Abstract and Discussion that they found “no significant protective role of breastfeeding… in the development of nonhereditary RB.” It would seem more accurate to say no association was observed between breastfeeding and severity of disease as measured by …..

Thank you for your comment. We have adjusted the language in the relevant sections of the paper which now read –

Line 87 (abstract) “This study suggests that breastfeeding neither impacts the sporadic development nor is associated with a decrease in the severity of non-hereditary RB as measured by age at diagnosis, stage of disease, ocular prognosis and extraocular spread”

and

“In this multicenter, multinational analysis of patients with RB, there was no significant association observed between breastfeeding and the severity of non-hereditary RB development (Line 296).”

  1. From the original Lag study, what proportion had the necessary data (i.e., completed the neonatal section)? What factors were associated with completion of this section?

Thank you for your question. The inclusion criteria were met in 654 of 692 patients who were included in the Lag Time study (94.5%). The section was completed if the physician was able to elicit a complete feeding history from the patient. This information is listed on line 208 of Results.

  1. How long were patients followed to determine whether they had died? Whereas other analyses pertain to characteristics around the time of diagnosis or shortly after (e.g., enucleation), mortality requires patient follow-up. More information about follow-up methods and completeness is needed to understand this analysis.

Thank you for your question. Patients were followed for approximately 1 year during the study to determine if they died. It is possible that events such as metastasis or death would have occurred after conclusion of the study. We have added a line in the limitations to reflect this: Patients were followed for one year, so there is a possibility that death or metastasis could have occurred after the conclusion of the follow-up period (Line 355).

  1. For some factors (e.g., lymph node involvement) there is considerable missing data. Can the authors discuss the potential implications of the missing data?

Thank you for your comment. We have added the following line in the paragraph discussing limitations to the study: While this is one of the largest multinational clinical databases on retinoblastoma, some patient reports were incomplete and lacked data in certain categories (such as lymph node involvement) which may also impact the analysis (Line 352).

Additional comments:

  1. How strong is the evidence from previous literature regarding perinatal and nutritional factors in terms of RB etiology? In its current wording, the introduction seems to suggest that the evidence is quite strong, but when I reviewed the publications cited in this section, it does not seem so clear.

Thank you for your comment. While certain nutritional factors have been studied in terms of RB etiology, the study of breastfeeding vs. formula feeding is quite lacking. We agree that the evidence from previous literature is not clear; In the introduction, we include “The hereditary genetic mechanisms underlying RB tumorigenesis are well characterized, but the role of perinatal nutrition and parental health factors in the etiology of developing new, somatic mutations are incompletely understood and the current literature on the subject is conflicting” to frame our discussion (Line 111).

  1. Please clarify how misclassification of hereditary status due to missing family history information could confound the study results (page 8, line 327).

Thank you for your question. Missing family history information could potentially allow for familial cases to be included within our analysis of sporadic RB since we were unaware of their genetic predisposition; while the majority of germline or familial RB cases are bilateral (tumors in both eyes), this is not always the case – we have excluded all bilateral patients however there is a small chance that a unilateral patient could have a germline mutation that we are unaware of. The course of RB development in these children would be greatly affected by their genetic background and it would be difficult to tease out the effect of feeding vs. their genetic predisposition.